# Trainee Evaluations of Preparedness for Clinical Trials in Medical Oncology—A National Questionnaire

**Michela Febbraro** [1,2,*] ⓘ**, Ghazaleh Kazemi** [2,3]**, Rosalyn Juergens** [2,3] **and Gregory R. Pond** [2,3]

1    Algoma District Cancer Program, Sault Ste Marie, ON P6B 0A8, Canada
2    Faculty of Health Sciences, McMaster University, Hamilton, ON L8S 4L8, Canada
3    Department of Oncology, Juravinski Cancer Center, Hamilton, ON L8V 5C2, Canada
*     Correspondence: michela.febbraro@medportal.ca

**Abstract:** Background: A standardized approach to the education of clinical trial investigators across Canadian medical oncology (MO) subspecialty training does not exist. With training programs transitioning to competency-based medical education (CBME), studies assessing education practices and competence are paramount to enhancing trainee education. This study aimed to determine whether current education practices in MO subspecialty training programs in Canada prepare trainees for participating in clinical trials as an investigator. Methods: From November 2021 to February 2022 a national, bilingual, online questionnaire to understand trainee experiences with self-perceived competence, preparedness, and willingness to participate in clinical trials as investigators was conducted. MO trainees, fellows, and new-to-practice physicians who completed an MO subspecialty training program in Canada were included. Results: A total of 41 responses were received (response rate: 15%). Formal training in how to participate in clinical trials as an investigator was reported by 73% of respondents. At the end of training, 65% of respondents rated competence in clinical trials as fair/poor and 74% rated preparedness in conducting clinical trials as fair/poor. Correlation analysis determined that in-clinic teaching in clinical trials trended toward improved self-evaluations of competence and preparedness ($p > 0.05$). Conclusion: This is the first study in Canada to assess competencies in any residency training program since the establishment of CBME. Training in conducting clinical trials is highly variable across MO programs in Canada, with most trainees finding current practices not translating into self-perceived competence and preparedness. Further assessment into how to produce competent clinical trial investigators is warranted.

**Keywords:** competency based medical education; clinical trials; residency

## 1. Introduction

Clinical trials are available in every discipline of medicine and have improved the standards of optimal patient care. Through the development of innovative treatments and the expansion of diagnostic techniques, cancer clinical trials become instrumental in identifying more effective outcomes and/or fewer adverse events for patients with cancer. In addition, the availability of clinical trials allows patients with cancer to access novel treatments that may not otherwise be available [1–5].

Despite the benefits of clinical trials, global estimates show that 3–8% of patients with a diagnosis of cancer enroll in clinical trials [6–8]. In Canada, clinical trial patient participation rates vary by region, with adult participation ranging from less than 1% in Newfoundland and Prince Edward Island, to approximately 6% in Alberta and Ontario [9]. There has been a push for increasing clinical trial patient enrollment at not only academic but community cancer centres across the country. Decentralized clinical trial delivery approaches aimed at engaging community cancer centres at a national level are currently being undertaken [10]. This increased engagement and recruitment for clinical trials will necessitate all medical oncologists, regardless of practice location, to be competent clinical trial investigators.

Several studies have examined reasons for poor clinical trial patient recruitment [5,7]. Prior investigator experience with and involvement in clinical trials during training can affect access and are positively associated with participation [5,11,12]. Given that investigator experience in clinical trials is correlated with increased clinical trial patient recruitment and patient participation, education literature suggests that the fundamentals of clinical trial design and clinical trial involvement earlier in a physician's career or during training should be adopted [4,11–13].

Historically, medical oncology subspecialty training was achieved through the traditional training model (Figure 1). In this model, a time-based approach occurred whereby trainees spent time in different rotations receiving evaluations determining successful completion. The curriculum succeeded in evaluating a trainee's knowledge with a minimal evaluation of competence or performance outside of knowledge-based domains. The achievement of competence was therefore assumed at the end of training and not rigorously evaluated [14,15]. In 2018, Canadian medical oncology subspecialty training programs switched to a competency-based medical education (CBME) model [16]. In this model, trainees progress through four stages of learning (Figure 2). CBME has four overarching themes: a focus on outcomes, an emphasis on abilities, a de-emphasis on time-based training, and the promotion of learner centeredness to produce competent physicians in both knowledge and skill-based domains [17]. The qualities and abilities a trainee should possess at the end of their training are not determined by the trainee but by national specialty committees and individual programs [16].

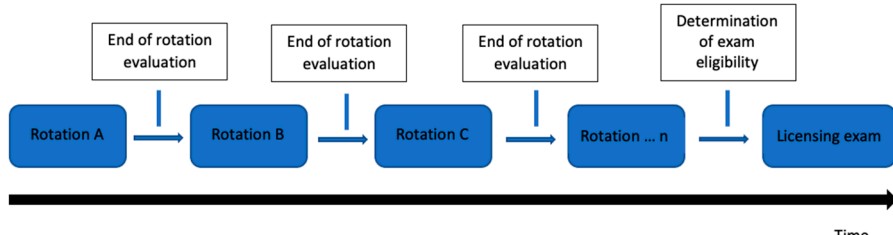

**Figure 1.** Traditional time-based training model in medical oncology. Trainees spend time in different rotations receiving evaluations determining successful completion within the subspecialty program.

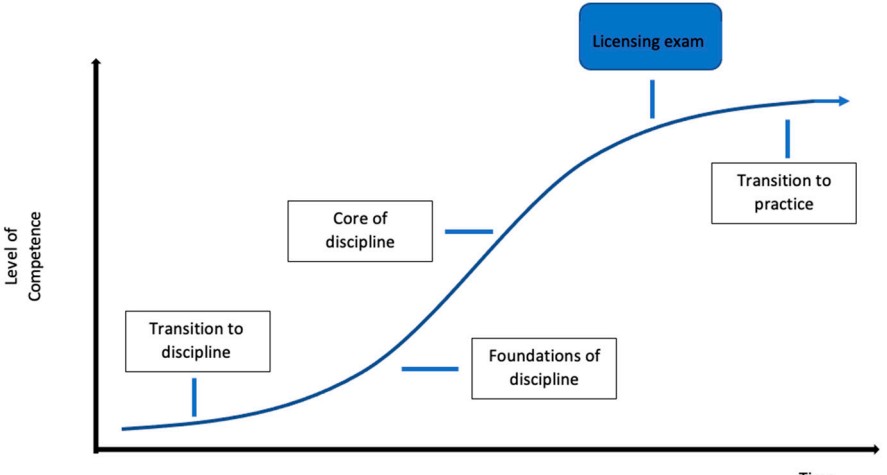

**Figure 2.** Competence-by-design training model in medical oncology. Trainees progress through four stages of training with the ability to matriculate through the stages with a focus on outcomes, an emphasis on abilities, and a de-emphasis on time-based training.

Despite the use of CBME as the training model in medical oncology and the importance of clinical trials in the field of medical oncology, there is no standardized formal training of clinical trial investigators in the field of medical oncology, and there are minimal

requirements to demonstrate competence [18–20]. Three competencies partially address clinical trial experience [21]. In the medical expert role, a trainee is required to establish a patient-centred management plan which can include a discussion of the available clinical trials. In the communicator role, the trainee must demonstrate their communication skills and strategies to help patients and their families make informed treatment decisions. This evaluation may include discussions regarding enrollment in clinical trials. In the scholar role, a trainee must contribute to work within a research program and demonstrate an awareness of clinical trials as a research tool. However, the trainee is not required to have been actively involved in or participated in the care of a clinical trial patient. Given the minimal training in conducting clinical trials within medical oncology subspecialty training programs, it is hypothesized that trainees may feel unprepared to participate in conducting clinical trials upon their graduation.

With medical oncology clinical trials becoming more accessible in Canada, there is a need to produce more competent clinical trial investigators. Currently, there are minimal requirements for training in the conduct of clinical trials. The purpose of this study is to evaluate whether current education practices in medical oncology subspecialty training prepare medical oncology trainees for participating in clinical trials as an investigator.

## 2. Materials and Methods

### 2.1. Study Design

This was a novel, web-based, bilingual questionnaire. Participants had to have completed training within five years and be a medical oncology subspecialty trainee, fellow, or new-to-practice physician. A trainee was defined as a student who is completing their core medical oncology subspecialty training (also known as a resident). A fellow was defined as a physician who has completed their medical oncology subspecialty training and had chosen to pursue additional training in medical oncology. A new-to-practice physician had to have completed their medical oncology training in Canada and be within their first five years of practice. In order to focus solely on medical oncology clinical trials, individuals who had completed their training outside of Canada or those in radiation oncology or surgical oncology training programs were excluded. Hematologic oncologists were excluded unless enrolled in a province whereby subspecialty training combines both medical oncology and hematologic oncology specialties within one training program (i.e., Quebec and British Columbia). An investigator of a clinical trial was defined as the physician responsible for the conduct of a clinical trial at a trial site [22]. Ethics approval was obtained through the Hamilton Integrated Research Ethics Board (HiREB).

### 2.2. Questionnaire Development

The Canadian Medical Association Journal (CMAJ) guide for the design and conduct of self-administered surveys was used to develop the questionnaire [23]. The questionnaire was developed to measure five domains: exposure to clinical trials and general research, self-perceived clinical trial competence, self-perceived preparedness to participate in clinical trials as an investigator, willingness to participate in clinical trials as an investigator, and the perceived role of the trainee to seek clinical trials experiences during training. An initial list of questions was generated through a combination of literature review and the modified Delphi technique involving experts and potential respondents [24]. This resulted in 100 items being generated. Items were reduced to eliminate redundancy and correspond to previously published Likert-scale anchors. The core questionnaire was narrowed to 17 items pertaining to the research question with the addition of eight demographic items.

The method for content validation was adapted from the ABC of content validation and content validity index calculation guide [25]. Six content experts were approached; four were identified based on being practicing oncologists with exposure to clinical trials and the medical oncology CBME curriculum, and two content experts were chosen given their expertise in questionnaire design and development. The content experts assessed the face validity of the overall questionnaire and the content validity of the individual

questionnaire items. The resulting questionnaire had a face validity of 0.94 and a content validity of 0.83 which satisfied the validation requirement.

The questionnaire was pre-tested with radiation oncology trainees at McMaster University in Hamilton, Ontario Canada. Radiation oncology trainees were chosen given that they work closely with medical oncology programs, often completing rotations in medical oncology, and are also exposed to the CBME curriculum. After pre-testing, the questionnaire was refined to improve flow and for the minimization of ambiguity. The final questionnaire consisted of 16 items on the research question with the addition of eight demographic items. The final instrument was translated into French.

### 2.3. Questionnaire Distribution

An electronic version of the final questionnaire was created on LimeSurvey for distribution. A call to participate in the study was sent via bilingual email to all Canadian medical oncology residency training program directors, program administrators, and fellowship coordinators. After the initial email invitation, two reminder emails were also sent. The Canadian Association of Medical Oncology (CAMO) also emailed the questionnaire to its entire membership. Responses were collected between November 2021 and February 2022. Respondents were not compensated for their participation.

### 2.4. Outcomes and Data Analysis

Descriptive statistics were used to summarize respondent characteristics and item responses. The self-assessed competence, preparedness, and willingness to participate in clinical trials were defined as co-primary endpoints. Outcomes were dichotomized into poor/fair versus good/very good/excellent for statistical purposes. Associations between a priori selected research questions and outcomes, as well as outcomes with each other, were assessed via Fisher's exact tests. Associations between outcomes were further evaluated using Spearman correlation coefficients based on response ranks. All tests were two-sided, and the statistical significance was defined at the $\alpha = 0.05$.

### 3. Results

Invitations to participate were sent to 360 CAMO members (all members that were medical oncologists regardless of years in practice) as well as 31 program directors, fellowship directors, and program assistants across Canada. Since the number of years of practice for CAMO members was not available, nor was the number of trainees at each site, the total number of eligible participants was not available, and a conservative estimate of 270 was used based on five-year data from the Canadian Resident Matching Service (CaRMS) [26]. A total of 41 eligible responses were received with a response rate of 15% (Table 1). Responses were evenly distributed between residents, fellows, and new-to-practice physicians. Most respondents (61%) were training or practicing in Ontario. Most new-to-practice physicians (64.7%) were practicing in an academic centre.

**Table 1.** Characteristics of respondents.

| Characteristics | | Responses—*N* (%) |
|---|---|---|
| *N* | | 41 |
| Participant status | Resident | 10 (24.4) |
| | Fellow | 14 (34.2) |
| | <5 years in practice | 17 (41.5) |
| Age | Mean (standard deviation) | 33 (27, 43) |
| Gender | Male | 13 (33.3) |
| | Female | 26 (66.7) |
| Location of medical oncology subspecialty program | Atlantic Canada | 3 (7.3) |
| | Quebec | 5 (12.2) |
| | Ontario | 25 (61.0) |
| | Central/Western Canada | 8 (19.5) |

**Table 1.** *Cont.*

| Characteristics | | Responses—*N* (%) |
|---|---|---|
| Practice setting * | Academic | 11 (64.7) |
| | Community | 4 (23.5) |
| | Mixed | 2 (11.8) |
| Graduate Level Degree (master's or PhD) | Yes | 16 (41.0) |
| | No | 23 (59.0) |
| Language of Questionnaire | English | 37 (90.2) |
| | French | 4 (9.3) |

* Practice setting was only asked for in-practice physicians (*N* = 17).

### 3.1. Exposure to Clinical Trials and General Research

Thirty-nine (95%) respondents noted they had experienced teaching regarding a clinical trial critical appraisal, with 30 (73%) indicating that teaching occurred in small group learning, followed by independent learning (Figure 3). This teaching was found to be adequate/very adequate by 24 (59%) respondents (Figure 4). Thirty-seven (90%) respondents indicated receiving teaching regarding clinical trial research methods, with 24 (59%) signalling that this teaching occurred in small-group learning, followed by in-clinic teaching. This teaching was found to be adequate/very adequate for 17 (41%) respondents. Conversely, 23 (56%) respondents did not receive teaching regarding participation in clinical trials. Regardless of teaching method (including no teaching), 28 (68%) respondents reported this training to be inadequate or very inadequate.

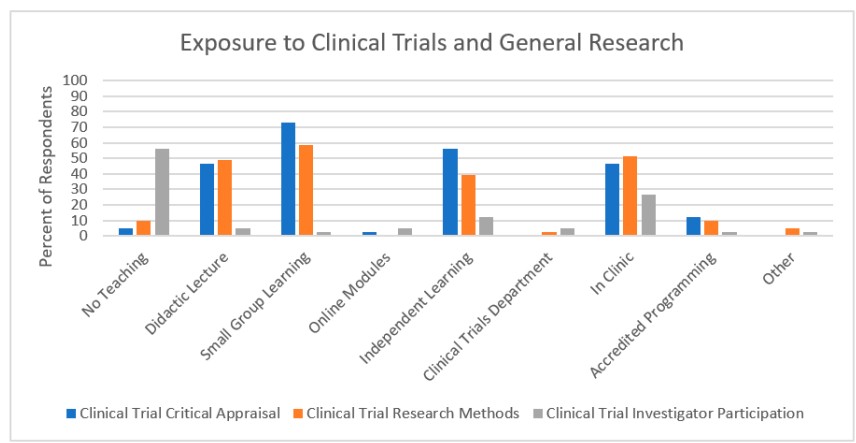

**Figure 3.** Results regarding types of clinical trial education received in medical oncology subspecialty training programs in Canada.

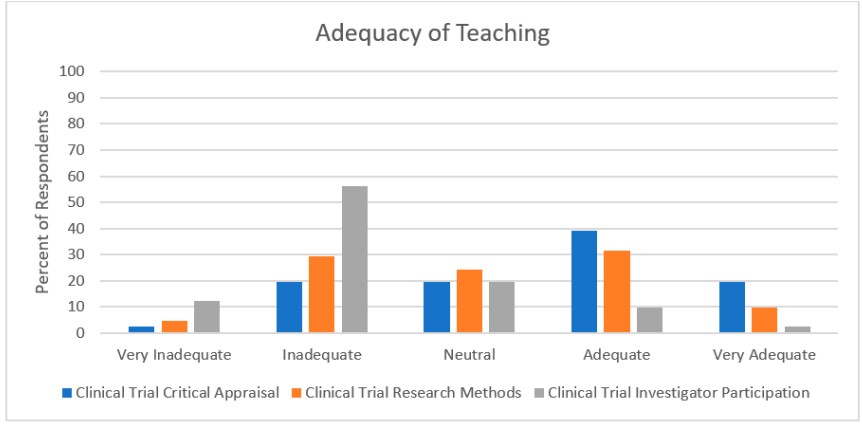

**Figure 4.** Results regarding the adequacy of clinical trial exposure and general research exposure in medical oncology subspecialty training programs in Canada.

*3.2. Self-Perceived Assessments*

Nineteen (65%) respondents rated their level of competence to participate in clinical trials as an investigator as poor/fair upon completion of training (Figure 5). Most respondents felt competent in tasks not directly related to clinical trial participation, such as searching for clinical trials (*n* = 24, 78%), referring a patient for a clinical trial (*n* = 28, 90%), or discussing clinical trials as a potential therapeutic option (*n* = 29, 94%) (Figure 6). A minority of respondents felt competent in more active roles within a clinical trial. Similarly, 29 (74%) respondents rated their level of preparedness to participate in clinical trials as an investigator as poor/fair upon completion of training. Conversely, 37 (95%) respondents rated their willingness to participate in clinical trials as an investigator as good/excellent.

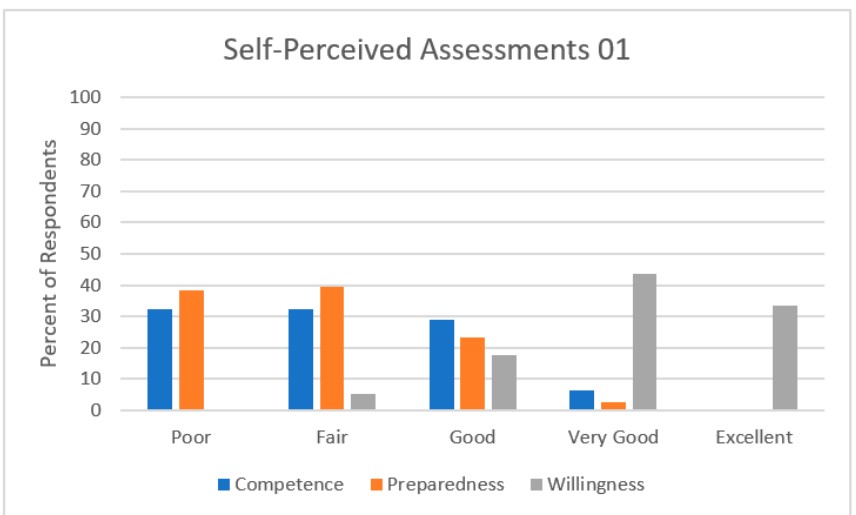

**Figure 5.** Self-perceived assessments of the level of competence, level of preparedness, and level of willingness to participate in clinical trials as an investigator at the completion of medical oncology subspecialty training in Canada.

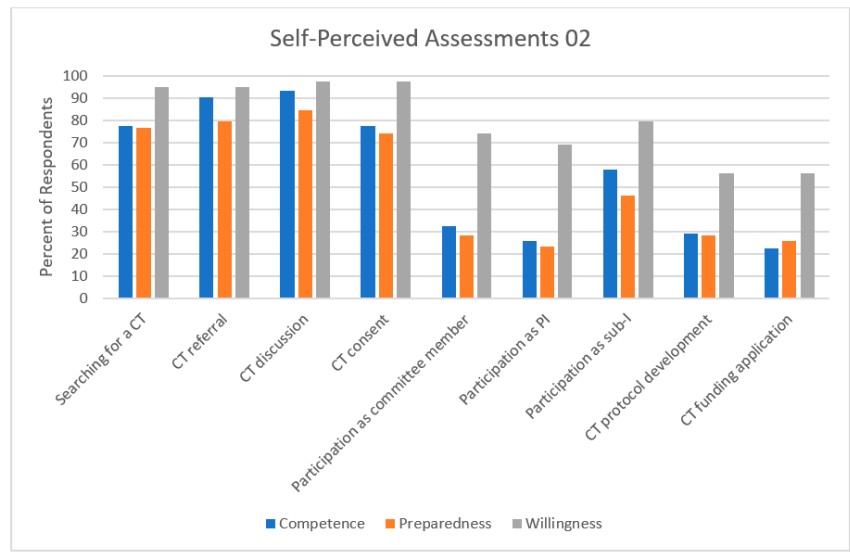

**Figure 6.** Self-perceived assessments of preparedness, willingness, and competence to participation in different aspects of clinical trials as a physician. CT—clinical trial; PI—principal investigator; sub-I—sub-investigator.

In the exploratory analysis to examine the association of competence with clinical trial training (Table 2), any teaching on how to participate in clinical trials was associated with a non-statistically significant improvement in competence (no teaching *n* = 4, 21% versus

any teaching $n = 7$, 58%, $p = 0.056$). There was a strong relationship between the level of competence and the level of preparedness for clinical trials (Spearman $\rho = 0.71$, $p < 0.001$).

**Table 2.** Associations of competence.

| Question | Answer * | N | Good to Excellent Competence $n$ (%) | *p*-Value |
|---|---|---|---|---|
| In your medical oncology subspecialty, how were you taught about participating in a clinical trial? | There was no teaching | 19 | 4 (21.1) | |
| | Any teaching present | 12 | 7 (58.3) | 0.056 |
| | In clinic teaching * | 7 | 4 (57.1) | |
| | All other teaching ** | 24 | 7 (29.2) | 0.21 |
| In your medical oncology subspecialty, how were you taught about critically appraising a clinical trial? | There was no teaching | 1 | 0 (0) | |
| | Any teaching present | 30 | 11 (36.7) | 1.00 |
| | In clinic teaching * | 7 | 4 (57.1) | |
| | All other teaching ** | 24 | 7 (29.2) | 0.21 |
| In your medical oncology subspecialty, how were you taught about clinical trials research methods? | There was no teaching | 3 | 1 (33.3) | |
| | Any teaching present | 28 | 10 (35.7) | 1.00 |
| | In clinic teaching * | 16 | 7 (43.7) | |
| | All other teaching ** | 15 | 4 (26.7) | 0.46 |
| Are you completing or have you completed a graduate level degree? | No | 16 | 6 (37.5) | |
| | Yes | 13 | 4 (30.7) | 1.00 |
| Upon completion of medical oncology subspecialty training, will you/did you feel the need to seek out additional clinical trials training? | No | 13 | 5 (38.5) | |
| | Yes | 16 | 5 (31.3) | 0.71 |

* Including clinical trials departmental teaching; ** including responses of no teaching.

### 3.3. Perceived Role of the Trainee

29 (74%) respondents felt it was very/extremely important to have a structured clinical trial curriculum. There was high agreement ($n = 33$, 85% and $n = 32$, 82% respectively) that it is the responsibility of the training program to ensure adequate clinical trial education and that this education prepares trainees for participation in clinical trials as an investigator. The role of the trainee to seek experiences autonomously in clinical trials was mixed with 18 (46%) of respondents noting it was the role of the trainee to seek experiences in clinical trials offered during training.

## 4. Discussion

This is the first study in Canada to assess clinical competencies in any residency training program since the establishment of CBME from the view of the trainee. Much of the literature concerning CBME design, planning, and oversight has been developed from the perspective of educators and administrators, with the resident voice involved in the assessment of transitioning to CBME [14,27–29]. There are no studies to date assessing competency within CBME graduates and the implemented learner milestones and entrustable professional activities (EPAs) created. The integration of the perspectives and provision of feedback upon CBME program implementation is necessary to ensure that user needs are being met [29].

Competencies, milestones, and EPAs should reflect future clinician practices. In Canada, there are 85 centres registered to perform clinical trials through the Canadian Cancer Trials Group (CCTG) with 35 of these centres designated as 'community centres' [30]. Oncologists, regardless of academic or community centre affiliation, are involved in designing, conducting, and participating in clinical trials. This is important considering the distribution of Canadian citizens, with approximately one-third of the population residing outside of urban centres where academic cancer centres are located [31]. National organizations are working to make trials more available to patients in community-based

centres to broaden treatment options for individuals in rural and underserved regions [10]. This study demonstrated that across Canada, current clinical trial education practices revolve around teaching critical appraisal and research methods with minimal teaching regarding how to participate in clinical trials as an investigator. The current clinical trial education practices in Canada resulted in respondents not feeling competent or prepared to participate in clinical trials as investigators upon completion of training. The rapid development of new drugs and therapeutic strategies in medical oncology has created an increase in the number of clinical trials available to patients. Access to clinical trials is complex, and there needs to be greater participation in clinical trials by physicians earlier in their careers. Medical oncology subspecialty trainees should receive education and training in the fundamentals of clinical trial design with a curriculum focused on how to participate in clinical trials as an investigator [13,32]. With early exposure to clinical trial research being positively associated with current participation in clinical research, an approach to clinical trial education that enhances the role integration of the subspecialty trainee to clinical trials and clinical research should be considered [11,12,32,33]. This can be done by balancing evidence-based medicine (EBM) with experiential learning. Standalone teaching in EBM, as is done currently in medical oncology curriculums, improves knowledge but does not improve skills, attitudes, or behaviours. Clinically integrated teaching in combination with an EBM curriculum can build knowledge and improve skills and competence while reinforcing positive attitudes toward clinical trials [34,35]. This study demonstrated that medical oncology subspecialty training programs offer an EBM curriculum, which builds knowledge. With CBME, there is an opportunity to provide a clinically integrated clinical trials program that builds on the current EBM curriculum, with the potential to improve trainee competence in clinical trials.

### *4.1. Limitations*

The response rate of 15% was lower than anticipated. The study made several attempts to optimize response rates in the study design via affiliation with CAMO, and multiple email outreach attempts. In addition, the questionnaire was developed in both English and French to increase equity. Direct outreach via travel and conferences was limited due to the coronavirus pandemic. Despite the low response rate, this study did demonstrate the current landscape for clinical trial education practices in Canada with representation from across the country.

Selection bias was introduced due to the use of a questionnaire-based study as they are more likely to appeal to potential respondents who are interested or engaged in the topic of the questionnaire. Most respondents were willing to participate in clinical trials and felt that there should be more clinical trial education in medical oncology subspecialty training. It is uncertain whether this directly reflects Canada's larger population of residents, fellows, and medical oncologists.

This study did not go beyond the level of the trainee. It did not assess educator or program director/administrator views. It did not assess productivity following graduation. This study aimed to obtain baseline information regarding clinical trial education in Canada as it relates to medical oncology. A qualitative participatory-action research approach may have been beneficial in conjunction with this questionnaire. This would allow further understanding of the importance of clinical trial education in training from the perspective of several stakeholders including residents and fellows. Clinical trials are important in every discipline of medicine; this questionnaire and subsequent mixed-methods assessments can be modified and used to assess trainee preparedness in other medical disciplines not only in Canada but internationally, with the possibility for its use in future reassessments. With modification and repeat assessments, reliability testing can be done longitudinally.

### *4.2. Special Considerations and Next Steps*

The importance of mentorship for the career growth and development of a clinical investigator is well-known and would have a substantial impact on CBME effectiveness [36].

Mentorship, also known as coaching, is at the forefront of CBME. The CBME coaching model involves both coaching in the moment and coaching over time [37]. Coaching over time involves a longitudinal coach/trainee relationship, which helps monitor a trainee's progress throughout CBME as they progress towards competence. The foundation of the coaching is based upon milestones and EPAs laid out within CBME. Without the prioritization of clinical trial training within CBME, coaching and mentorship in clinical trials may not occur and would be very difficult to standardize. This study did not assess the impact of mentorship as it relates to clinical trials in training as it was not part of the scope of this project. However, other studies in research and medical education have noted its importance in training [38–40].

The COVID-19 pandemic was a catalyst for the digital age. In medical education, there was a substantial shift in online learning, which allowed educational programs to be maintained [41]. Certain medical processes and procedures could not be digitally translated. This includes in-clinic and patient-facing experiences for trainees. The COVID-19 pandemic led to decreased research productivity and decreased clinical trial enrollment [42]. The impact of the disruption on teaching activities and clinical learning remains unknown [43]. Given the decreased clinical trial activity and reduced in-person interactions, it is likely that the COVID-19 pandemic influenced clinical trial education. It is unclear if the observed lack of self-perceived competence and preparedness by medical oncology trainees will persist as the impact of the COVID-19 pandemic recedes.

Current CBME competencies addressing clinical trials should change such that they reflect the requirements to create competent clinical trial clinicians within the field of medical oncology. Before this study, there were no data on the outcomes of current clinical trial education practices in medical oncology subspecialty training. With this study, a need has been identified in medical oncology. However, clinical trials are important in other areas of medicine, and CBME is becoming internationally renowned and globally adopted [44,45]. Further information will be necessary to propose to current CBME leaders the necessity of strengthening the clinical trial curriculum, not only for medical oncology programs in Canada but internationally. This will require collaboration between CBME leaders, program directors, program education leaders, and trainees.

## 5. Conclusions

Clinical trials are a foundational component in the field of medical oncology. This study demonstrates that trainees and physicians in medical oncology are willing to participate in conducting clinical trials but do not perceive themselves as competent or prepared to do so upon completion of training. Medical oncology subspecialty programs should consider a focus on experiential learning and modifying CBME requirements for clinical trial education to effectively prepare trainees to become competent clinical trial clinicians.

**Author Contributions:** Conceptualization, M.F., R.J., G.K. and G.R.P.; methodology, M.F. and G.R.P.; validation, M.F.; formal analysis, M.F. and G.R.P.; writing—original draft preparation, M.F.; writing—review and editing, M.F., R.J., G.K. and G.R.P.; supervision, G.R.P. All authors have read and agreed to the published version of the manuscript.

**Funding:** This research received no external funding.

**Institutional Review Board Statement:** The study was conducted in accordance with the Declaration of Helsinki and approved by the Institutional Review Board of McMaster University (protocol code 14088 and date of approval 15 October 2021).

**Informed Consent Statement:** Informed consent was obtained from all subjects involved in the study.

**Data Availability Statement:** The data presented in this study are available upon request from the corresponding author.

**Conflicts of Interest:** The authors declare no conflict of interest.

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
