# Peer review of "Trainee Evaluations of Preparedness for Clinical Trials in Medical Oncology—A National Questionnaire"

_curroncol, doi:10.3390/curroncol30080553_

Round 1

Reviewer 1 Report

Eloquent research and manuscript. As recognized by the authors- the only drawback is the poor response rate to survey (noted as 15% but perhaps even lower as survey was sent out to more participants).  Selection bias brings into question the validity of conclusions, nonetheless, these limitations have been addressed by the authors. 

Author Response

We thank the reviewer for taking the time to review our manuscript. We recognize the limitation of the limited response rate of only 15%. As the reviewer indicates, this brings into concern potential issues regarding selection bias. Unfortunately, poor response rate is a common problem with many surveys. We have highlighted this issue within the limitations section and the attempts which were made to increase the response rate.

Attached you will find the revised copy of the manuscript with all reviewers' suggestions and comments taken into consideration. 

Reviewer 2 Report

The authors conducted a survey of medical oncology trainees in Canada and found that most trainees did not feel that they received sufficient training as competent investigators to conduct clinical trials. A major problem that requires revision in this paper is the ambiguous use of the word "participate" which could apply to patients enrolled in a clinical trial. Please see attached pdf file for specific comments.

Author Response

Thank you for taking the time to review the manuscript. We appreciate the nuances between participating and conducting clinical trials and are appreciative that you highlighted areas whereby the text was ambiguous. We have made modifications throughout the manuscript by revising the wording from ‘participating in clinical trials’ to ‘participating in the conduct of clinical trials. 

Attached you will find the updated manuscript taking into account all comments and suggestions from the reviewers. 

Reviewer 3 Report

Febbraro et al. conducted a questionnaire-based study aimed at evaluating the preparedness of medical oncology trainees and new-to-practice physicians for clinical trials. This novel study holds crucial implications for enhancing the training of medical oncologists and improving patient recruitment to clinical trials in Canada. It's worth noting that the participation rate was low, approximately 14%, resulting in a limited pool of only 41 responses, as indicated by the authors as well. As a consequence, the findings should be interpreted with caution due to the small sample size.

To strengthen the presentation of their results, the authors are strongly encouraged to use graphs rather than just tables. This visual representation would enhance the clarity and impact of their most significant data points.

Minor issues:

·       In line 14, clarification is needed on the time frame: "From November 2021 to February"; did the authors intend to cover the period from Nov 2021 to Feb 2022?

·       In line 17, the term "completed a MO subspecialty" should be revised to "completed an MO subspeciality" for grammatical accuracy.

Author Response

Thank you for taking the time to review the manuscript. We have added figures 3-6 to improve the clarity of the manuscript as suggested. The typographical errors were also addressed.

Attached you will find the revised manuscript taking into account all comments/suggestions from reviewers. 

Round 2

Reviewer 2 Report

The authors have adequately addressed issues raised about ambiguity in the text.

Reviewer 3 Report

All modifications acknowledged. Thanks.